# Market of Stocks during Crisis Looks Like a Flock of Birds

**DOI:** 10.3390/e22091038

**Published:** 2020-09-17

**Authors:** Bahar Afsharizand, Pooya H. Chaghoei, Amirhossein A. Kordbacheh, Andrey Trufanov, Golamreza Jafari

**Affiliations:** 1Department of Physics, Iran University of Science and Technology, Tehran 512, Iran; afsharizand@gmail.com (B.A.); akordbacheh@iust.ac.ir (A.A.K.); 2Center for Complex Networks & Social DataScience, Shahid Beheshti University, Tehran 512, Iran; pooya1991@gmail.com; 3Institute of Information Technology and Data Science, Irkutsk National Research Technical University, 664000 Irkutsk, Russia; troufan@gmail.com; 4Department of Physics, Shahid Beheshti University, Tehran 512, Iran; 5Department of Network and Data Science, Central European University, 1007 Budapest, Hungary

**Keywords:** collective behavior, Vicsek model, financial crisis

## Abstract

A crisis in financial markets can be considered as a collective behaviour phenomenon. The collective behaviour is a complex behaviour which exists among a group of animals. The Vicsek model has been adapted to represent this complexity. A unique phase space has been introduced to represent all possible results of the model. The return of the transaction volumes versus the return of the closed price of each share has been used within the defined phase space. The findings show that the direction of the resultant velocity vectors of all share in this phase space act in the same direction when the financial crisis happens. By monitoring the market’s collective behaviour, it will be possible to gain more knowledge about the condition of the market days in crisis. This research aims to investigate the collective behaviour of stocks using the Vicsek model to study the condition of the market during the days in crisis.

## 1. Introduction

The emergence of collective behaviour has been observed in many systems, such as schools of fish or flocks of birds. One of these systems’ main properties is that the members move in the same direction without any leadership [1,2,3]. This behaviour exists to increase the chances of survival for the whole system. This behaviour can also be seen in financial stock markets. Independent firms are exposed to more business risk than similar firms that are in a joint group. Independent markets are more vulnerable and have fewer chances of survival in a financial crisis. Despite growing stronger being in a group for each member, it would itself be a major factor to spread a problem of each member within the whole group. One of the most important issues in businesses is the prediction of this significant shift during a financial crisis [4,5,6,7,8]. This has been considered in previous studies with a variety of approaches [9,10]. Computer-aided platforms have been used in the past to facilitate decision-making in financial markets. For example, high-frequency trading is carried out by powerful computers to analyse markets and buy or sell shares within seconds [11]. These platforms assume that the share price changes are fixed across various financial markets. This assumption is not valid if the market copes with financial crises. This is the main motivation of applying collective behaviour of birds and animals to the stock markets [12]. In order to consider the collective behaviour of stock markets, particle swarm optimisation (PSO) has been adapted in [13]. The algorithm was focused on choosing the parameters of trading strategy to predict price trends and to obtain the highest net return. Malekian et al. have shown that PSO and ant colony approaches are more effective than regression to predict financial crashes [14].

Thomas and Francisco [15] applied the collective behaviour of animals to the financial markets and used network theory to show the complexity of their acts. In science, significant efforts have been made to identify and understand the general principles governing collective behaviours [16,17,18]. These efforts have been unsuccessful due to the absence of equilibrium in such systems. The interactions between the particles of these systems are short-range. Therefore, the existing statistical physics techniques used for the long-range interaction of the particles are unable to express the system’s dynamics. The Vicsek model (VM) has been applied to construct a framework for this study. The Vicsek model was first introduced [19] in 1995 to investigate the emergence of self-ordered motion in systems of particles with biologically motivated interaction. This model was further extended in 1999 [20] to demonstrate that a system of self-propelled particles (SPP) exhibits spontaneous symmetry breaking and self-organisation in one dimension. In 2000, Levin et al. proposed a model [21] in which he enhanced the scope of Vicsek by taking into account both attractive and repulsive forces. The Buhl model [22] (2006), which implies a modification of velocity weighting of the particles, approves rapid transition from the disordered movement of individual agents to highly aligned collective motion. In contrast to the formless and point-like particles described within the Vicsek model, Peruani attributed forms and packing coefficients to the agents [23]. There are many more models based upon Vicsek platforms such as Chate [24], Cocker-Smiley [25], Galam [26], and that of the work in [27]. The Vicsek model deals with a system of self-propelled particles. The basic idea in the Vicsek model is that at each time step, a given particle moves with a constant velocity. A new direction is obtained by the average direction of all the neighbourhood particles within a circle of the radius with including a random noise [28]. In other words, all the particles of the system tend to follow their neighbours’ direction. However, the particles might change their directions while following their neighbours’ trajectory, which is considered noise. Vicsek model has been proven to be effective for modelling collective behaviour with the ability to express such systems’ dynamic states. Thus, the Vicsek model has been adopted to build the framework for this research. The dynamics of the stock market is a very complex problem. We used VM to take some snapshots at different times of a market to compare them. Thus, the Vicsek model has been adopted to build the framework for this research. Neither studying dynamics of the Vicsek model nor the stock market dynamics has not been considered.

## 2. The Validation of the Framework

To validate the proposed framework, the historical data of the $Standard\&Poor^{\prime }s500\;\left(S\&P500\right)$ companies in the US stock market has been used. The S&P500 index, an index of the New York Stock Exchange, consists of the 500 largest companies in the US. It is a market value-weighted index (stock price times number of shares outstanding), with each stock’s weight in the index proportionate to its market value. The S&P500 index is one of the most widely used benchmarks of US equity performance [29]. Our data covers 12 years (from January 2007 to February 2019) with a recording frequency of daily intervals. The data gathered from 465 stocks includes closing prices and transaction volumes.

### Prototype Implementation

VM assumes that particles move in a two-dimensional plane with periodic boundary conditions. In this model, each particle’s location is defined by X-Y coordinates, and a vector can obtain its velocity in the plane. The model assumes that the particles move with equal speed and the direction of the velocity vectors of particles changes as a function of time. The direction of each particle’s motion depends on the average velocity, ${\vec{V}}_{n}\left(t\right)$, of its neighbouring particles within a circle of radius $r_{0}$. The model defines a circular neighbourhood area $U_{n}\left(r_{0},t\right)$ centered at the n-th particle position ${\vec{r}}_{n}\left(t\right)$. The number of closed neighbours is shown as $K_{n}\left(t\right)$. According to the VM model, no interactions can occur for the n-th particle located outside the $U_{n}\left(r_{0},t\right)$. The average velocity of the neighbours of the particle *n* is calculated by:(1)$${\vec{V}}_{n}\left(t\right)=\frac{1}{K_{n}\left(t\right)}\sum_{j:{\vec{r}}_{j}\in U_{n}}{\vec{\nu }}_{j}\left(t\right),$$
where ${\vec{\nu }}_{j}\left(t\right)$ denotes the velocity of j-th particle inside $U_{n}\left(r_{0},t\right)$. The Equation (Equation 1) is the sum of the velocities of each particle in the neighbourhood divided by the number of particles in that neighbourhood.

Equation (Equation 2) shows the time evolution of the velocity angel of n-th particle,
(2)$$\theta_{n}\left(t+\Delta t\right)=Angle\left[{\vec{V}}_{n}\left(t\right)\right]+\eta \xi_{n}\left(t\right),$$
where $\xi_{n}\left(t\right)$ is a random variable uniformly distributed in the interval [−π,π], and η is the noise intensity, which is a positive parameter.

Therefore, the velocity of n-th particle can be calculated by Equation (Equation 3)
(3)$${\vec{\nu }}_{n}\left(t+\Delta t\right)=\nu_{0}Cos\left(\theta_{n}\left(t+\Delta t\right)\right)\hat{x}+\nu_{0}Sin\left(\theta_{n}\left(t+\Delta t\right)\right)\hat{y},$$
where $\hat{x}$ and $\hat{y}$ denote X and Y direction respectively. Then, the position of n-th particle in the next time step is:(4)$${\vec{r}}_{n}\left(t+\Delta t\right)={\vec{r}}_{n}\left(t\right)+{\vec{\nu }}_{n}\left(t+\Delta t\right)\Delta t.$$

At any time *t*, the magnitude of order in a system might be assessed through the order parameter $\Phi_{\eta }\left(t\right)$ [30,31], which is expressed by:(5)$$\Phi_{\eta }\left(t\right)=\frac{1}{N\nu_{0}}\left|\sum_{n=1}^{N}{\vec{\nu }}_{n}\left(t\right)\right|,$$
where the η indicates that the order parameter in the system depends on the noise intensity. Equation (Equation 5) is the magnitude of the sum of all the particle’s velocity in the system divided by the total number of particles *N* by the value of the velocity $\nu_{0}$. This parameter must be in the interval [0, 1]; 0 if the velocity vectors are distributed randomly, and 1 if particles line up in the same direction.

To use the Vicsek model, a coordinate system is required to describe the movements of stocks in a market. Thus, X-Y values have been formulated as follows,
(6)$$X_{t}=P_{t}-P_{t-1}\;\;\;\;\;\;\;\;\;\;\&\;\;\;\;\;\;\;\;\;\;Y_{t}=T_{t}-T_{t-1}\;\;,$$
where *t* takes just discrete values 1,…,t−1,t,t+1,…, $P_{t}$ is the closing price, and $T_{t}$ is the transaction volume at time *t*. The X-axis and the Y-axis of the coordinate system show the daily closed price difference and the difference in transaction volumes for two consecutive days. Based on VM, the density of particles has been considered as the main factor in collective behaviour. This research does not take into account how the market trend changes over a time period. Thus, the effect of the market trend on each stock has been neutralised using the following transformations,
(7)$$x_{t}=\left(X_{t}-\mu_{x}\right)/\sigma_{x}\;\;\;\;\;\;\;\;\;\&\;\;\;\;\;\;\;\;\;\;y_{t}=\left(Y_{t}-\mu_{y}\right)/\sigma_{y}\;\;,$$
where $\mu_{x}$: moving average over 20 days, $\mu_{y}$: moving average over 20 days and $\sigma_{x}$: moving standard deviation over 20 days and $\sigma_{x}$: moving standard deviation over 20 days.

The new phase space can be generated from the above formula. The speed parameter for a daily stock can be calculated as follow,
(8)$${\vec{\nu }}_{t}=\left(x_{t+1}-x_{t}\right){\hat{s}}_{x}+\left(y_{t+1}-y_{t}\right){\hat{s}}_{y}\;\;,$$
where ${\hat{s}}_{x}$ and ${\hat{s}}_{y}$ are unit vectors along X-axis and Y-axis, respectively.

## 3. Results and Discussion

Figure 1 represents the velocity vectors of all one-day stocks for three critical market days compared to the market’s those of normal functioning of the market. Figure 1 compares the velocity vectors for normal days in the market (non-crisis) versus markets in crisis over three days for 465 stocks. It has been observed that during the non-crisis period, the stocks tend to move randomly. The direction of vectors has been highlighted with different colours in Figure 1. The direction of vectors in Figure 1 for markets in crisis shows the emergence of the collective behaviours.

To differentiate the normal days from the days in crisis in the stock market, a value has been assigned to each day. The following steps are used to calculate this value.
Drawing an imaginary circle and lay the start point of the velocity vectors of each share $\nu_{i}\left(t\right)$ on the center of the circle.Dividing the circle into 12 equal slices and assigning a number to each of them, so each velocity vector is located in one slice.Mapping each share to one slice number. For example, the velocity vector of i-th share in the division l is shown as ${\vec{\nu }}_{l}^{i}\left(t\right)$, where *t* denotes time. *l* can only accept integer values and it returns the number of the slice that velocity vector is located, and Δθ is the degree of each slice which in this research assumed 30 degrees.
(9)$$l=\left\lfloor \theta_{i}/\Delta \theta \right\rfloor \;\;.$$
*l* equals to the integer part of the right hand side of Equation (Equation 9).Coupling the joint shares. For example, if share i is in slice l and share j is in slice m, the return of the joint l-m or m-l will be considered. l-m and m-l are considered as one possible combination of joint shares. Therefore, the joint probability of both i and j to be placed in l and m divisions $p({\vec{\nu }}_{l}^{i}\left(t\right),{\vec{\nu }}_{m}^{j}\left(t\right))$ equals 1.Counting the possible combinations of joint shares $C_{t}$. For example, 0-0, 0-1, 0-2, ..., 10-11, 11-11. It can be a maximum of 78 combinations of joint shares.The distribution of these possible combinations of joint shares $P_{t}\left(l,m\right)$ can be obtained using Equation (Equation 10). The calculated $P_{t}\left(l,m\right)$ values for normal days vs. days in crises have been presented in Figure 2C.
(10)$$P_{t}\left(l,m\right)=\sum_{i,j\in 1,\ldots ,N}p\left({\vec{\nu }}_{l}^{i}\left(t\right),{\vec{\nu }}_{m}^{j}\left(t\right)\right).$$

Number of possible combinations of joint shares have been depicted in Figure 2B. The Y-axis values have been calculated using Equation (Equation 11).
(11)$$C_{t}=\sum_{l,m\in 0,1,\ldots ,11}Sign\left(P_{t}\left(l,m\right)\right).$$
In Equation (Equation 11), $P_{t}\left(l,m\right)$ only accepts non-negative values. When $P_{t}\left(l,m\right)$ equals to zero, $Sign(P_{t}\left(l,m\right))$ returns zero. When $P_{t}\left(l,m\right)$ is greater than 0, $Sign(P_{t}\left(l,m\right))$ returns 1. With this arrangement, the number of possible joint shares can be obtained.

On a normal day in the market, 78 possible combinations of joint shares (l, m) can be found. This number can be significantly lower on a day in crisis. When the number is lower than 78, it also indicates that collective behaviour is emerging in the market. It means for some values of l and m, $P_{t}\left(l,m\right)$ equals zero. For each day, the number of l and m combination where $P_{t}\left(l,m\right)$ is greater than zero is counted. This number has been estimated for the trades happened all open days of 2018 and shown in the middle chart in Figure 2. We used the reference index of S&P500 to compare the calculated value from this research with the values listed in normal days in the market. This comparison has been depicted in Figure 2. Moreover, the distribution of possible combinations between joint shares in normal days has been compared with the same distribution during the market crisis (Figure 2, the bottom chart).

Comparing the data from normal days against days in crisis in Figure 2 shows that the numbers of possible combinations for joint shares have decreased significantly during the days in crisis.

This research also illustrates how a time delay in the stock market can affect the emergence of collective behaviour. In Figure 3, the emergence of collective behaviour has been depicted for the duration of 8 days. The delay of one day has been assumed in calculations. To measure the alignment of the two velocity vectors, a value has been defined.
The velocity vector of i-th share at time t is shown by ${\vec{\nu }}_{i}\left(t\right)$.The alignment of the velocity vectors of two separate shares can be calculated by Formula (Equation 12). The basis of this calculation is the cosine of the angle between the two velocity vectors.
(12)$$a_{ij}\left(t\right)=\frac{\nu_{i}\left(t\right).\nu_{j}\left(t\right)}{|\nu_{i}\left(t\right)||\nu_{j}\left(t\right)|}.$$The stocks normally are correlated to each other. Thus, the change in one stock can affect the other stocks. This effect does not happen at once. Thus, we considered a delay for the calculations.
(13)$${\tilde{a}}_{ij}\left(t\right)=Max\left(\frac{\nu_{i}\left(t\right).\nu_{j}\left(t-\tau \right)}{|\nu_{i}\left(t\right)||\nu_{j}\left(t-\tau \right)|},\frac{\nu_{i}\left(t\right).\nu_{j}\left(t\right)}{|\nu_{i}\left(t\right)||\nu_{j}\left(t\right)|},\frac{\nu_{i}\left(t-\tau \right).\nu_{j}\left(t\right)}{|\nu_{i}\left(t-\tau \right)||\nu_{j}\left(t\right)|}\right).$$
Equation (Equation 13) selects the maximum alignment factor between velocity vectors of two shares for two consecutive days. τ which is the added lag for alignment matrix calculations has been assumed 1 day.The alignment matrix has been constructed using calculated ${\tilde{a}}_{ij}\left(t\right)$ in Equation (Equation 13). The alignment matrix has been generated using dendrogram to present the stocks which are grouped together.

According to Figure 3, All stocks in the market are globally aligned and move in the same direction. This pattern did not occur at once. It started on 21 December 2018, spread on 24 December 2018 [32] and all the stocks in the market were involved on this date. On 26 December 2018, the collective behaviour started to disappear. On 27 December 2018, the market shows normal behaviour. This proves that monitoring the collective behaviour in the stock market can be useful. This is mainly because it can warn the traders about the crisis ahead.

## 4. Conclusions

The results of empirical research show that the stock market during crisis days follows collective behaviour patterns. Days in crisis are affected by the price of shares and transaction volume fluctuations. Thus, it is crucial to define a phase space based on each share price and volume. The direction of the defined velocity vectors within the phase space can be used to identify the days in crisis. This phenomenon can be seen in the behaviour of biological systems, such as groups of animals. The Vicsek model has been used previously to investigate the collective behaviour in these systems. Suggested framework has shown the potential to represent this behaviour in the stock markets as well.

Raw data from S&P500 have been collected between 2007 and 2019. The collective behaviour has been discovered in some days in crisis. During these days, the direction of velocity vectors has been aligned as it was expected. An order parameter has been defined to compare the collective behaviour of the market during the crisis against normal days in the market. Using the data in S&P500, 24th of December 2018 has been identified as a crashed day. It is concluded that by analysing the market data before this date, it will be possible to identify the potential crisis in the days ahead.

## Figures and Tables

**Figure 1 entropy-22-01038-f001:**
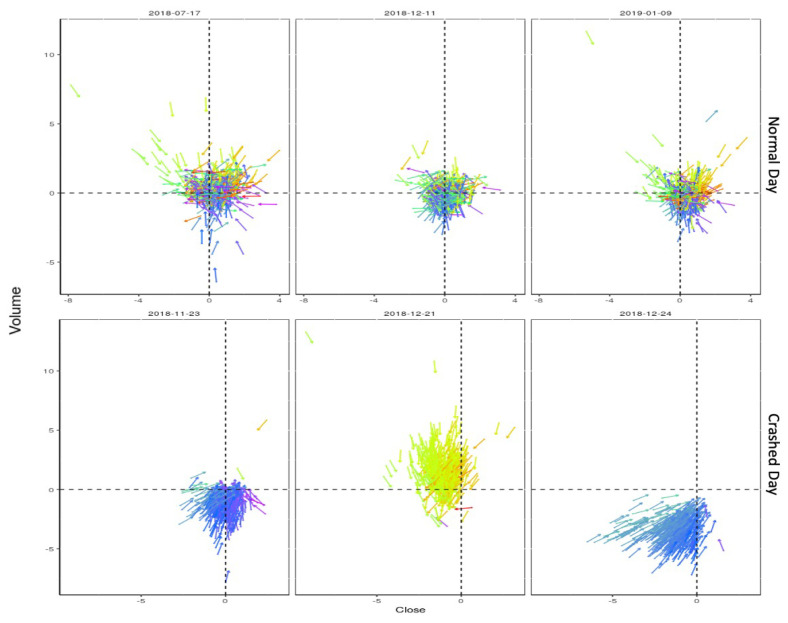
Collective behaviour of stocks in crisis. Comparison of stock movement for three critical days and for those away from crisis. For each share in the stock market, only the direction of the velocity vector has been considered in this research.

**Figure 2 entropy-22-01038-f002:**
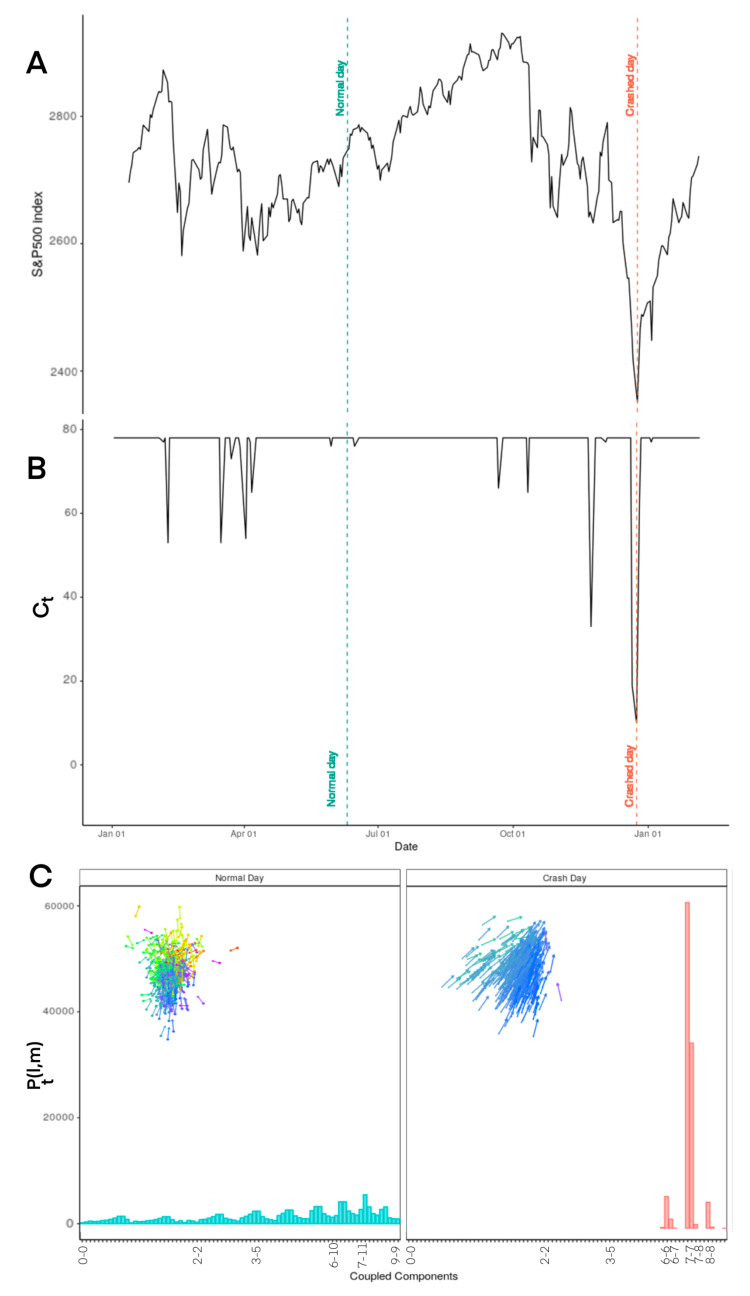
Comparing normal day and crisis for S&P500 including 468 companies. (**A**) The historical data of S&P500 for 2018 (all days). (**B**) The number of possible combinations of joint shares $C_{t}$ in 2018 (normal day vs. days in crisis). (**C**) Distribution of possible combinations of joint shares $P_{t}\left(l,m\right)$ in 2018 (normal day vs. days in crisis).

**Figure 3 entropy-22-01038-f003:**
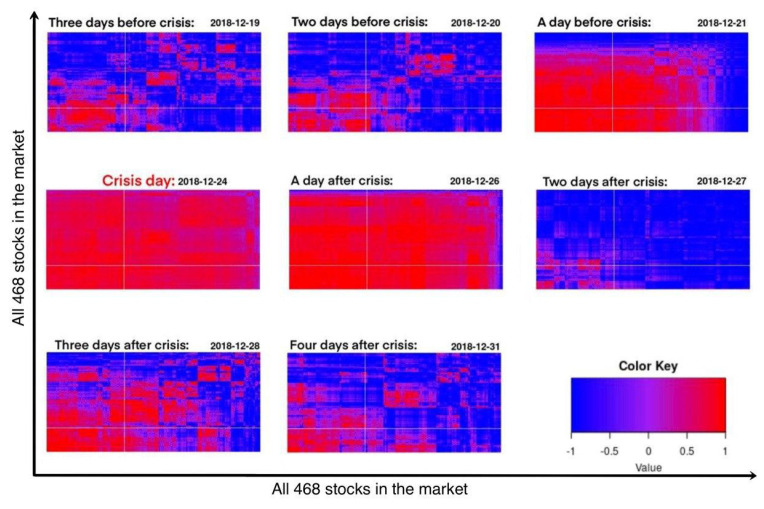
The maximal alignment of pairs of stock vectors, when compared on the same day or with a one day lag.This plot has been generated using dendrogram and aligned shares have been grouped together. This graph shows that during the days in crisis one giant group of shares is shaped. This can be shown by a solid red color.

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
