# Peer review of "Market of Stocks during Crisis Looks Like a Flock of Birds"

_entropy, 2020, doi:10.3390/e22091038_

Round 1

Reviewer 1 Report

It is difficult to convey how disappointed I felt about the lack of attention given by the authors to the manuscript revision before submission. The overall impression is it was written in a rush and carelessly: there are many misprints and some sentences do not have sense or seem to be misplaced. See for instance lines 28-29,  lines 32-34, lines 140-146 and the list may continue.

I find this attitude disrespectful to the reviewer, who is not a spell checker or a proofreader. 

I ask the authors to rewrite the manuscript and to resubmit the paper. 

I would be happy to review and judge the scientific content of a new, extensively edited, version of the paper.

Author Response

In the new version of the manuscript, we have improved writing issues and thanks to the reviewers for their thorough work which has inspired the authors greatly.

Reviewer 2 Report

The article by Afsharizand et al. constructs a two-dimensional phase space for stocks, with dimensions of closing price and trading volume.  The authors show that the velocity vectors in this space are mostly aligned in market crises.   The phase-space construction and the observed vector alignment during crises are both interesting.  However, the current presentation of the work is not acceptable for publication.  

The most obvious problem is that the writing, mathematics, and presentation style are all sloppy.  This gives the impression that the paper was thrown together without a second reading.  It is certainly not in a state ready for external review.  There are many glaring oversights that the authors could have easily identified themselves that must be rectified.  I will mention some of these later in my detailed comments.

  Besides the untidiness of the presentation, there are also serious conceptual issues with the work.  For example, the comparison of the present work with the Vicsek model for the dynamics of flocking is misleading.  In particular, the Vicsek model is never actually used in the paper, except perhaps as motivation to define the variables and order parameter.  However, to emphasize, the Vicsek model is a model of dynamics.  And the autonomous dynamics of the Vicsek model are most certainly not representative of stock market dynamics.     The introduction would be clearer if it noted that the stock market is not an autonomous dynamic but, rather, responds to external influences.  Whereas birds and fish etc. can flock without any obvious driving, this paper shows that the stock market exhibits flocking only during times of crisis.   Can this apparent interdependence among stocks in crisis be explained by their common driving?  I.e., are stock dynamics conditionally independent given external driving?  Perhaps approximately.  But if the external driving is unknown, then there may appear to be effective interactions among the stocks during times of strong driving.  These effective interactions may lead to an ephemeral appearance of flocking.   This of course requires a definition of flocking.  In the popular Vicsek model, flocking can be quantified by the mentioned order parameter.  The authors may choose to adopt this order parameter while explicitly acknowledging the distinct dynamics of the stock market.   The top panels of Fig. 1 seem to suggest that the primary dynamic on "normal" days is a regression toward the mean.   Perhaps the most promising aspect of the paper is represented in Fig. 2.  Unfortunately, this figure is currently difficult to interpret since the "coupled components" were not clearly introduced.   The authors speculate that observed flocking may be used to predict a crisis.  However, this speculation is dubious because the authors have shown that flocking only occurs in crises, rather than before them.   Indeed, what is a crisis other than a situation where many stocks plummet together?  In this sense, ephemeral flocking is the crisis.   If a causal order were imposed, it may plausibly be argued that the external crisis causes the ephemeral flocking rather than the converse.  No evidence was presented that flocking can anticipate a crisis.  Rather, it seems to be an indicator of currently being in a crisis.   All of the above comments should be considered seriously if the authors aim to revise their manuscript.  Ideally, this would allow the authors to better contextualize their work and extract further useful insights.     Besides the general comments above, the following list offers more detailed comments and suggestions:   The authors should make a serious attempt to improve the English in the paper.  Currently, there seems to be a grammatical error every few sentences, which distracts from the science.  E.g., just on the first page:   o  Line 18: "The question arises as to what causes the collective behavior of the stocks of the companies be?"    o  Line 21:  "causes accumulate of predators"   o  Line 22:  "in spite of getting stronger to be in a group for each member "   o  The sentence on Line 28 lacks a beginning.   Although the authors make a more concrete connection, aspects of stock market behavior have been compared to flocks of birds in previous publications.  E.g., in Gerig's "High-Frequency Trading Synchronizes Prices in Financial Markets".  An effort should be made to acknowledge this, and compare the current work with previous related works.   Page 2, Line 33: It is not clear what is meant by "over the latest 25". Perhaps "over the last 25 years"? The authors should clarify.   The section introducing the Vicsek model, like the rest of the paper, is a bit sloppy.   Velocity in this model is two-dimensional.  Obviously, a real-valued two-dimensional vector can be represented as the real and imaginary parts of a complex number, as in Eq. (3).  But the authors should be clear from the beginning about whether v_n(t) is represented as a two-dimensional vector or a complex number in their mathematical framework.   Eq. (5): The variable N is never introduced.  Supposedly, this is the total number of agents (i.e., stocks).     \nu_0 is supposedly a real-valued constant representing the constant speed of all agents.  The paper should say so explicitly.   Eq. (7): Different notation should be used to distinguish X_t and Y_t on the left vs. right-hand side of the equations.  E.g., via primes or tildes. Eq. (7) is currently written as one would assign a variable in computer code; but it is not a mathematically appropriate way to describe the change of variables. Similarly, Eq. (6) and (8) should refer to the appropriate variable names, rather than each referring to different interpretations of X_t and Y_t.   There is a further conflict of notation in Eqs. (6) and (8). In Eq. (6), V_t is the trading volume. In Eq. (8), V_t is the velocity vector for the stock. Distinct variable names should be used for these distinct concepts.   The colors of the arrows in the figures should be explained.   Are all lengths of the arrows the same in the figures? If so, it should be mentioned that only the velocity direction (rather than the velocity vector defined by Eq. (8)) is shown in the figures.   Pair sectors are not introduced clearly.  It is very difficult to understand what a "pair sector" is from the description around Line 123. The way that I interpreted it, it becomes very difficult to reconcile Figs. 1 and 2. Perhaps a graphic demonstrating the partitioning would be useful.   Line 129: "Stock space is severely restricted" when the pairs are distributed over all of the space?  I do not understand this statement.    Fig. 2 seems very interesting.  Unfortunately, it is difficult to understand since the pair sectors were not clearly introduced.   Lines 143 to 145 seems to paraphrase the preceding paragraph.  Choose one.  

Author Response

Reviewer 2:

The article by Afsharizand et al. constructs a two-dimensional phase space for stocks, with dimensions of closing price and trading volume.  The authors show that the velocity vectors in this space are mostly aligned in market crises.   The phase-space construction and the observed vector alignment during crises are both interesting.  However, the current presentation of the work is not acceptable for publication.  

The most obvious problem is that the writing, mathematics, and presentation style are all sloppy.  This gives the impression that the paper was thrown together without a second reading.  It is certainly not in a state ready for external review.  There are many glaring oversights that the authors could have easily identified themselves that must be rectified.  I will mention some of these later in my detailed comments. Besides the untidiness of the presentation, there are also serious conceptual issues with the work.  For example, the comparison of the present work with the Vicsek model for the dynamics of flocking is misleading.  In particular, the Vicsek model is never actually used in the paper, except perhaps as motivation to define the variables and order parameter.

We should appreciate referee with constructive comments and careful reading.

VM studies the collective behaviour of birds in spatial coordinates using the alignment order parameter for velocity vectors. The same approach has been used to research the collective behaviour of stocks in the stock market. The new coordinate space has been defined which includes the return of the closed price and the return of the transaction volume during the days in crisis. The order parameter is the pair coupling between stocks vectors, which we have introduced in Eq. 9.

Line 80-81: To use the Vicsek model, a coordinate system is required to describe the movements of stocks in a market.

Line 126-133: In Fig 3, the emergence of collective behaviour has been depicted for the duration of 8 days. The delay of one day has been assumed in calculations. To estimate the alignment of the velocity vectors, the cosine of the angle between two velocity vectors has been calculated. According to Fig 3, All stocks in the market are globally aligned and move in the same direction. This pattern did not occur at once. It started on 21/12/2018, spread on 24/12/2018 [28] and all the stocks in the market were involved on this date. On 26/12/2018, collective behaviour started to disappear. On 27/12/2018, the market shows normal behaviour. This proves that monitoring collective behaviour in the stock market can be useful. This is mainly because it can warn the traders about the crisis ahead.

Line 139-141: This phenomenon can be seen in the behaviour of biological systems such as groups of animals. Viscek model has been used previously to investigate the collective behaviour in these systems. VM has shown the potential to represent this behaviour in the stock markets as well.

However, to emphasize, the Vicsek model is a model of dynamics.  And the autonomous dynamics of the Vicsek model are most certainly not representative of stock market dynamics. The introduction would be clearer if it noted that the stock market is not an autonomous dynamic but, rather, responds to external influences.  Whereas birds and fish etc. can flock without any obvious driving, this paper shows that the stock market exhibits flocking only during times of crisis.   Can this apparent interdependence among stocks in crisis be explained by their common driving?  I.e., are stock dynamics conditionally independent given external driving?  Perhaps approximately.  But if the external driving is unknown, then there may appear to be effective interactions among the stocks during times of strong driving.

Self-propelled particles also referred to as self-driven particles, are terms used by physicists to describe autonomous agents, which convert energy from the environment into directed or persistent motion. Birds and fishes normally do not show consistent behaviour in nature. However, when an external factor threads them such as a shark tries to hunt fishes, suddenly their pattern changes to collective behaviour as a defence strategy. Viscek focuses on modelling these types of behaviours. Also, the effect of this behaviour on neighbours markets in finance is a complex issue. To reduce this complexity, it is necessary to investigate the relations between stocks. Fig 3 presents that the behaviour mentioned above has emerged before the market crashed. 

 These effective interactions may lead to an ephemeral appearance of flocking.   This of course requires a definition of flocking.  In the popular Vicsek model, flocking can be quantified by the mentioned order parameter. The authors may choose to adopt this order parameter while explicitly acknowledging the distinct dynamics of the stock market. The top panels of Fig. 1 seem to suggest that the primary dynamic on "normal" days is a regression toward the mean. Perhaps the most promising aspect of the paper is represented in Fig. 2.  Unfortunately, this figure is currently difficult to interpret since the "coupled components" were not clearly introduced. 

Line 101-113: The following steps are used to calculate this value:

  1. Drawing an imaginary circle around the velocity vectors of each stock.
  2. Dividing the circle into some equal slices (our case is 12 equal slices) and assigning a number to each of them, so each velocity vector is located in one slice.
  3. Mapping each stock to one slice number.
  4. Coupling the joint shares, (VA,VB). For example, if share VA is in slice 4 and share VB is in slice 6, the return of the joint 4-6 or 6-4 will be considered. 4-6 and 6-4 is considered as one possible combination of joint shares.
  5. Counting the possible combinations of joint shares, P(VA,VB). For example, 1-1,1-2,1-3,...,11-12,12-12. It can be a maximum of 78 combinations of joint shares.

  The authors speculate that observed flocking may be used to predict a crisis.  However, this speculation is dubious because the authors have shown that flocking only occurs in crises, rather than before them.   Indeed, what is a crisis other than a situation where many stocks plummet together?  In this sense, ephemeral flocking is the crisis.   If a causal order were imposed, it may plausibly be argued that the external crisis causes the ephemeral flocking rather than the converse.  No evidence was presented that flocking can anticipate a crisis.  Rather, it seems to be an indicator of currently being in a crisis. 

Line 122-129: In Fig 3, the emergence of collective behaviour has been depicted for the duration

of 8 days. The delay of one day has been assumed in calculations. To estimate the alignment of the

velocity vectors, the cosine of the angle between two velocity vectors has been calculated. According

to Fig 3, All stocks in the market are globally aligned and move in the same direction. This pattern did

not occur at once. It started on 21/12/2018, spread on 24/12/2018 and all the stocks in the market were

involved on this date. On 26/12/2018, collective behaviour started to disappear. On 27/12/2018,

the market shows normal behaviour. This proves that monitoring collective behaviour in the stock

market can be useful. This is mainly because it can warn the traders about the crisis ahead.

All of the above comments should be considered seriously if the authors aim to revise their manuscript.  Ideally, this would allow the authors to better contextualize their work and extract further useful insights.   Besides the general comments above, the following list offers more detailed comments and suggestions:   The authors should make a serious attempt to improve the English in the paper.  Currently, there seems to be a grammatical error every few sentences, which distracts from the science.  E.g., just on the first page:   o  Line 18: "The question arises as to what causes the collective behavior of the stocks of the companies be?"    o  Line 21:  "causes accumulate of predators"   o  Line 22:  "in spite of getting stronger to be in a group for each member "   o  The sentence on Line 28 lacks a beginning.  Although the authors make a more concrete connection, aspects of stock market behavior have been compared to flocks of birds in previous publications.  E.g., in Gerig's "High-Frequency Trading Synchronizes Prices in Financial Markets".  An effort should be made to acknowledge this, and compare the current work with previous related works. 

Modern ICT platforms facilitate financial firms to buy and sell stocks with high performance to reach thousands of trades a second [Mandes, A. (2016) Algorithmic and high-frequency trading strategies: A literature review. Working Paper . MAGKS Joint Discussion Paper Series in Economics, No. 25-2016, Philipps-University Marburg, School of Business and Economics, Marburg ]. The weird thing is that computer-aided “high-frequency” trading (HFT) makes price changes similar ( in other words  “synchronized”) for all the corporations across financial markets. And it has been attractive to compare stocks with birds and fish in their synchronous behaviour—Gerig in his [Gerig, A. (2012). High-Frequency Trading Synchronizes Prices in Financial Markets. SSRN Electronic Journal. 10.2139/ssrn.2173247] represented effectively and collated price trajectories of financial securities and motion of schooling fish thus to infer analogy of asserts to animal groups. 

Based on the cooperation principle a particle swarm optimization (PSO) algorithm was considered by [Crawford, B. , Soto, R., Martín, M., de la Fuente-Mella, H., Castro, C. & Paredes, F.. (2018). Automatic High-Frequency Trading: An Application to Emerging Chilean Stock Market. Scientific Programming. 2018. 1-12. 10.1155/2018/8721246. ] to design a fully automatic HFT system. The algorithm was focused on choosing the parameters of trading strategy to predict price trends and to obtain the highest net return with no concern to the situation on the market in general. The paper  [Malekian E., Fakhari, H.  Qasemi , J.   Farzad, S.  A Comparative Study of Prediction Stock Crash Risk by using Meta-Heuristic & Regression. International  Journal of Finance and Managerial Accounting, Vol.3, No.9, Spring 2018. 63-77 ] asserted that the  PSO and ant colony approaches are more effective than traditional regression one to predict financial crashes.

 Page 2, Line 33: It is not clear what is meant by "over the latest 25". Perhaps "over the last 25 years"? The authors should clarify.  

Due to changes in the new version, we omit this sentence.

 The section introducing the Vicsek model, like the rest of the paper, is a bit sloppy.  Velocity in this model is two-dimensional.  Obviously, a real-valued two-dimensional vector can be represented as the real and imaginary parts of a complex number, as in Eq. (3).  But the authors should be clear from the beginning about whether v_n(t) is represented as a two-dimensional vector or a complex number in their mathematical framework.

In the new version, we changed formulas and assigned a vector notation to them like equation 1,2,3,4,... 

Eq. (5): The variable N is never introduced.  Supposedly, this is the total number of agents (i.e., stocks). \nu_0 is supposedly a real-valued constant representing the constant speed of all agents.  The paper should say so explicitly. 

Line 76-77: Equation 5 is the absolute value of the sum of all the particle's velocity in the system divided by the total number of particles N by the value of the velocity ν_{0}.

   Eq. (7): Different notation should be used to distinguish X_t and Y_t on the left vs. right-hand side of the equations.  E.g., via primes or tildes. Eq. (7) is currently written as one would assign a variable in computer code; but it is not a mathematically appropriate way to describe the change of variables. Similarly, Eq. (6) and (8) should refer to the appropriate variable names, rather than each referring to different interpretations of X_t and Y_t.   There is a further conflict of notation in Eqs. (6) and (8). In Eq. (6), V_t is the trading volume. In Eq. (8), V_t is the velocity vector for the stock. Distinct variable names should be used for these distinct concepts. 

Equation 6,7,8 are changed in the new version and use different notation.

 The colors of the arrows in the figures should be explained.  

Line 97-98: The direction of vectors has been highlighted with different colours in Fig 1.

 Are all lengths of the arrows the same in the figures? If so, it should be mentioned that only the velocity direction (rather than the velocity vector defined by Eq. (8)) is shown in the figures. 

In caption figure 1: For each share in the stock market, only the direction of the velocity vector has been considered in this research. 

Pair sectors are not introduced clearly.  It is very difficult to understand what a "pair sector" is from the description around Line 123. The way that I interpreted it, it becomes very difficult to reconcile Figs. 1 and 2. Perhaps a graphic demonstrating the partitioning would be useful.

Line 129: "Stock space is severely restricted" when the pairs are distributed over all of the space?  I do not understand this statement.    Fig. 2 seems very interesting.  Unfortunately, it is difficult to understand since the pair sectors were not clearly introduced.   Lines 143 to 145 seems to paraphrase the preceding paragraph.  Choose one.  

Line 123-124: Comparing the data from normal days against days in crisis in Fig 2 shows that the numbers of possible combinations for joint shares have decreased significantly during the days in crisis.

Reviewer 3 Report

The manuscript under review applies the Vicsek model to the dynamics of stock markets, finding that during financial crises stocks behave as a synchronized bird flock. I found the paper interesting and well written, and the conclusions appear convincing. 

Nevertheless, I have found some small issues I would like the Authors could address before I can endorse the work for publication. In particular, I wonder if the interconnections among stocks which make the collective dynamics emerge in crises work also during economic booms or rebounds: also in a positive trend I would expect coherent trajectories in the phase space. Could the authors write some considerations about that?

Secondly, I do not understand why the authors did not show the behaviour of the order parameter defined in Equation (5), or in Equation (9). They claim "one might use a scalar parameter...", but then they do not use it. I think that a figure showing the transition during a crisis by means of such parameter would be very useful in make their conclusions clearer and stronger.

Minor point: page 3, line 89, there is a typo: "Regarding this closing prices price..." 

After the authors address the few points I have raised above, I will be happy to give my definitive endorsment to the paper.

Author Response

Reviewer 3:

The manuscript under review applies the Vicsek model to the dynamics of stock markets, finding that during financial crises stocks behave as a synchronized bird flock. I found the paper interesting and well written, and the conclusions appear convincing. 

Nevertheless, I have found some small issues I would like the Authors could address before I can endorse the work for publication. In particular, I wonder if the interconnections among stocks which make the collective dynamics emerge in crises work also during economic booms or rebounds: also in a positive trend I would expect coherent trajectories in the phase space. Could the authors write some considerations about that?

Secondly, I do not understand why the authors did not show the behaviour of the order parameter defined in Equation (5), or in Equation (9). They claim "one might use a scalar parameter...", but then they do not use it. I think that a figure showing the transition during a crisis by means of such parameter would be very useful in make their conclusions clearer and stronger.

Minor point: page 3, line 89, there is a typo: "Regarding this closing prices price..." 

After the authors address the few points I have raised above, I will be happy to give my definitive endorsment to the paper.

We appreciate the referee for a positive comment. We have considered her/his comment in the manuscript.

Line 9-10: This research aims to investigate the collective behaviour of stocks using the Viscek model to study the condition of the market during the days in crisis. Thus, the boundary of the research limits to the market study during the days in crisis. 

The summary, VM studies the collective behaviour of birds in spatial coordinates using the alignment order parameter for velocity vectors. The same approach has been used to research the collective behaviour of stocks in the stock market. The new coordinate space has been defined which includes the return of the closed price and the return of the transaction volume during the days in crisis. The order parameter is the pair coupling between stocks vectors, which we have introduced in Eq. 9.

Also, by detrending the closed price and transaction volume, we simplified the issue and removed the other limitations such as positive or negative market trends.  

Round 2

Reviewer 1 Report

In my opinion scientific content of the manuscript is sufficient. 

Main criticisms are: 1) Vicsek model describes the dynamics of a group of particles. However the present manuscript is limited to detecting a kind of synchronization phenomenon but I do not see any analysis of the dynamics of the stock market as compared to the Vicsek model.  I deem the sentence ''VM  has shown the potential to represent this behaviour in the stock markets as well.'' an overstatement. Is VM only a motivation for their analysis? I think the manuscript will benefit from some comments on this point. 2) I cannot understand formula (9) and how it relates to steps from 1 to 6. Please explain. 3) Axis labels in Figure2 and Figure 3 cannot be read. Captions are not sufficiently clear as they not contain enough information to understand the figures.

Finally, I appreciate authors' efforts to improve the writing of the manuscript -much better than the previous version - but the presentation can still be improved: many typos and poor phrasing here and there. I think the editor should decide if the presentation is sufficiently good to achieve the journal standards.

Author Response

reviewer1:

In my opinion scientific content of the manuscript is sufficient. 

Main criticisms are: 1) Vicsek model describes the dynamics of a group of particles. However the present manuscript is limited to detecting a kind of synchronization phenomenon but I do not see any analysis of the dynamics of the stock market as compared to the Vicsek model.  I deem the sentence ''VM  has shown the potential to represent this behaviour in the stock markets as well.'' an overstatement. Is VM only a motivation for their analysis? I think the manuscript will benefit from some comments on this point. 

The dynamics of the Vicsek model has not been considered in this paper. We have used the Vicsek model to model collective behaviour among the stocks, although they differ in some aspects. Stock markets have their dynamics. Using the Vicsek model the dynamics of the stock market has been studied. The main focus of this study is to compare the collective behaviour of the stock market when it is in crisis vs. the normal days in the market. We agree with the reviewer that the dynamics of the stock market has not been simulated. 

2) I cannot understand formula (9) and how it relates to steps from 1 to 6. Please explain. 

We have explained the steps in more detail and expanded the formulas.

3) Axis labels in Figure2 and Figure 3 cannot be read. Captions are not sufficiently clear as they not contain enough information to understand the figures.

The labels and captions of figures are edited as they can be more readable.

Finally, I appreciate authors' efforts to improve the writing of the manuscript -much better than the previous version - but the presentation can still be improved: many typos and poor phrasing here and there. I think the editor should decide if the presentation is sufficiently good to achieve the journal standards.

We should appreciate the referee help improve the paper with her/his constructive comments.

Reviewer 2 Report

2nd Round of Comments on Afsharizand et al.'s "Market of stocks during crisis looks like a flock of birds".

The revised manuscript improves on the previous version. In particular, the definition of joint shares is now more clear.
However, further improvements are still necessary.

In some places, the revisions introduced new problems. For example, the introduction was revised; but the first sentence is no longer a sentence.
The new Eq. (9) seems promising, but it does not make sense as is.

Other problems identified in the first version of the paper were never addressed. Most significantly, the revision never addressed my previous major concern:
This work indeed uses a phase-space model inspired by the Vicsek model; however it does not use the Vicsek dynamics at all. This distinction should be made clear.

I again emphasize that the Vicsek model is a model of dynamics.
The authors construct a comparable phase space, and they show the alignment of the stocks' velocity vectors. This is all analogous to an analysis that could be done for the Vicsek model or any 2-D flocking model. But it does not mean that the dynamics of the stock market emulates the dynamics of the Vicsek model. It appears that the authors have used the Vicsek model to inspire an analysis of stock flocking. It is fine to discuss the Vicsek model. But it should be made clear that Vicsek dynamics were not simulated nor otherwise analyzed in this work. Even a simple statement about this would help to align the reader's expectations.

Other previous concerns were also not addressed sufficiently. For example, Eq. (3) is still not technically correct. Most readers will know what was intended---that the components of the velocity vector correspond to the real and imaginary parts of the complex-valued expression on the right hand side. But it would be nice if the presentation was technically correct.

Eq. (9) should be remedied.
The summand is never defined.
N is a number, not a set.
Also, it is not clear how to interpret the right-hand side of Eq. (9) since we are told that \ell is supposed to be integer-valued. However, \theta is a continuous-valued variable.
My understanding is that \ell is a range of angles, rather than a particular angle.

Line 127:
"The delay of one day has been assumed in calculations."
It is not clear what calculations are being discussed, nor how delay could enter the calculations. Please clarify.

The caption for Fig. 3 should be more clear about what is being shown. Supposedly it is the velocity alignment between select pairs (all pairs?) of stocks.

In their response to my comments, the authors give a nice literature review of relevant work in flocking dynamics in trading.
They state:

"Modern ICT platforms facilitate financial firms to buy and sell stocks with high performance to reach thousands of trades a second [Mandes, A. (2016) Algorithmic and high-frequency trading strategies: A literature review. Working Paper . MAGKS Joint Discussion Paper Series in Economics, No. 25-2016, Philipps-University Marburg, School of Business and Economics, Marburg ]. The weird thing is that computer-aided “high-frequency” trading (HFT) makes price changes similar ( in other words “synchronized”) for all the corporations across financial markets. And it has been attractive to compare stocks with birds and fish in their synchronous behaviour—Gerig in his [Gerig, A. (2012). High-Frequency Trading Synchronizes Prices in Financial Markets. SSRN Electronic Journal. 10.2139/ssrn.2173247] represented effectively and collated price trajectories of financial securities and motion of schooling fish thus to infer analogy of asserts to animal groups."

"Based on the cooperation principle a particle swarm optimization (PSO) algorithm was considered by [Crawford, B. , Soto, R., Martín, M., de la Fuente-Mella, H., Castro, C. & Paredes, F.. (2018). Automatic High-Frequency Trading: An Application to Emerging Chilean Stock Market. Scientific Programming. 2018. 1-12. 10.1155/2018/8721246. ] to design a fully automatic HFT system. The algorithm was focused on choosing the parameters of trading strategy to predict price trends and to obtain the highest net return with no concern to the situation on the market in general. The paper [Malekian E., Fakhari, H. Qasemi , J. Farzad, S. A Comparative Study of Prediction Stock Crash Risk by using Meta-Heuristic & Regression. International Journal of Finance and Managerial Accounting, Vol.3, No.9, Spring 2018. 63-77 ] asserted that the PSO and ant colony approaches are more effective than traditional regression one to predict financial crashes."

This all seems like it would be relevant as part of the introduction, but it was never incorporated into the manuscript. Is there a good reason not to include this in the introduction? I would expect that it would be appropriate to include there.

Minor comments:

Line 76:
Should change `absolute value' to `magnitude'

Lines 94-95:
Check grammar.

Lines 102-111:
This adds a useful clarification.

Lines 130-133:
This is a useful addition to the paper, suggesting that there is indeed an opportunity to leverage this work for prediction.

Line 140:
Change "Viscek" to "Vicsek"

Author Response

Reviewer2:

The revised manuscript improves on the previous version. In particular, the definition of joint shares is now more clear.

However, further improvements are still necessary.

In some places, the revisions introduced new problems. For example, the introduction was revised; but the first sentence is no longer a sentence.

The new Eq. (9) seems promising, but it does not make sense as is.

 Updated version explained equation 9 in more detail.

Other problems identified in the first version of the paper were never addressed. Most significantly, the revision never addressed my previous major concern:

This work indeed uses a phase-space model inspired by the Vicsek model; however it does not use the Vicsek dynamics at all. This distinction should be made clear. I again emphasize that the Vicsek model is a model of dynamics. The authors construct a comparable phase space, and they show the alignment of the stocks' velocity vectors. This is all analogous to an analysis that could be done for the Vicsek model or any 2-D flocking model. But it does not mean that the dynamics of the stock market emulates the dynamics of the Vicsek model. It appears that the authors have used the Vicsek model to inspire an analysis of stock flocking. It is fine to discuss the Vicsek model. But it should be made clear that Vicsek dynamics were not simulated nor otherwise analyzed in this work. Even a simple statement about this would help to align the reader's expectations.

The dynamics of the Vicsek model has not been considered in this paper. The Vicsek model has been used to model collective behaviour among birds and stocks, although they differ in some aspects. Using the Vicsek model, the dynamics of the stock market has been studied. The main focus of this study is to compare the collective behaviour of the stock market when it is in crisis vs. the normal days in the market. The author agrees with the reviewer that the dynamics of the stock market has not been simulated. 

Other previous concerns were also not addressed sufficiently. For example, Eq. (3) is still not technically correct. Most readers will know what was intended---that the components of the velocity vector correspond to the real and imaginary parts of the complex-valued expression on the right hand side. But it would be nice if the presentation was technically correct.

 The author agrees with the reviewer about the misunderstanding that can arise from this formula. Thus, this formula has been replaced in the updated paper without any significant change in the core value of this research.

Eq. (9) should be remedied. The summand is never defined. N is a number, not a set.

It is updated.

Also, it is not clear how to interpret the right-hand side of Eq. (9) since we are told that \ell is supposed to be integer-valued. However, \theta is a continuous-valued variable.

My understanding is that \ell is a range of angles, rather than a particular angle.

In line 120 of the new version we mention it.

Line 127:

"The delay of one day has been assumed in calculations."

It is not clear what calculations are being discussed, nor how delay could enter the calculations. Please clarify.

In the new version, we clarify the delay calculation in steps in line 147-153. 

The caption for Fig. 3 should be more clear about what is being shown. Supposedly it is the velocity alignment between select pairs (all pairs?) of stocks.

It is for all pairs and the caption of figure 3 is updated.

In their response to my comments, the authors give a nice literature review of relevant work in flocking dynamics in trading.

They state:

"Modern ICT platforms facilitate financial firms to buy and sell stocks with high performance to reach thousands of trades a second [Mandes, A. (2016) Algorithmic and high-frequency trading strategies: A literature review. Working Paper . MAGKS Joint Discussion Paper Series in Economics, No. 25-2016, Philipps-University Marburg, School of Business and Economics, Marburg ]. The weird thing is that computer-aided “high-frequency” trading (HFT) makes price changes similar ( in other words “synchronized”) for all the corporations across financial markets. And it has been attractive to compare stocks with birds and fish in their synchronous behaviour—Gerig in his [Gerig, A. (2012). High-Frequency Trading Synchronizes Prices in Financial Markets. SSRN Electronic Journal. 10.2139/ssrn.2173247] represented effectively and collated price trajectories of financial securities and motion of schooling fish thus to infer analogy of asserts to animal groups."

"Based on the cooperation principle a particle swarm optimization (PSO) algorithm was considered by [Crawford, B. , Soto, R., Martín, M., de la Fuente-Mella, H., Castro, C. & Paredes, F.. (2018). Automatic High-Frequency Trading: An Application to Emerging Chilean Stock Market. Scientific Programming. 2018. 1-12. 10.1155/2018/8721246. ] to design a fully automatic HFT system. The algorithm was focused on choosing the parameters of trading strategy to predict price trends and to obtain the highest net return with no concern to the situation on the market in general. The paper [Malekian E., Fakhari, H. Qasemi , J. Farzad, S. A Comparative Study of Prediction Stock Crash Risk by using Meta-Heuristic & Regression. International Journal of Finance and Managerial Accounting, Vol.3, No.9, Spring 2018. 63-77 ] asserted that the PSO and ant colony approaches are more effective than traditional regression one to predict financial crashes."

This all seems like it would be relevant as part of the introduction, but it was never incorporated into the manuscript. Is there a good reason not to include this in the introduction? I would expect that it would be appropriate to include there.

As the referee suggestion, we have added this part to the introduction in lines 22-31.

Minor comments:

Line 76:

Should change `absolute value' to `magnitude'

 Done.

Lines 94-95:

Check grammar.

Done.

Lines 102-111:

This adds a useful clarification.

Lines 130-133:

This is a useful addition to the paper, suggesting that there is indeed an opportunity to leverage this work for prediction.

Line 140:

Change "Viscek" to "Vicsek"

Done.

We appreciate the referee with accurate comments. 

Reviewer 3 Report

After the revision, the manuscript can now be published.

Author Response

We should appreciate again the referee for helping improve the paper with her/his constructive comments.

Round 3

Reviewer 1 Report

Great improvements from the previous versions.  I recommend publication after the following comments has been addressed.

  • I ask the authors to state explicitly that neither dynamics of the Vicsek model nor the the dynamics of the stock market have not be considered  in their work.
  • 1) Line 22: Computer-aided platforms HAS BEEN used in the past to facilitate decision-making in financial markets. 
  • 2) Lines 30-31: “[…] the algorithm in [13] does not take  into account the situation of the market.” Which kind of situations may effect the results of algorithm in [13]? Please explain.
  • 3) Line 87: is THE magnitude.
  • 4) Line 123-126: I am sorry, but this part is not clear yet.  Conditional probability? Conditional on what? Is not that p is just an indicator function, that equals 1 when the i-th share and j-th share are in the l-th  and m-th slice respectively, otherwise equals zero? Please explain.

Author Response

We greatly appreciate the reviewer with her/his constructive comments. Really, we can not compare the new version with the first version of the manuscript.

Great improvements from the previous versions.  I recommend publication after the following comments has been addressed.

  • I ask the authors to state explicitly that neither dynamics of the Vicsek model nor the dynamics of the stock market have not be considered in their work.

     We added it in the lines 57-60.

  • 1) Line 22: Computer-aided platforms HAS BEEN used in the past to facilitate decision-making in financial markets. 

       done.

  • 2) Lines 30-31: “[…] the algorithm in [13] does not take  into account the situation of the market.” Which kind of situations may effect the results of algorithm in [13]? Please explain.

      We edited this sentence.

  • 3) Line 87: is THE magnitude.

        done.

  • 4) Line 123-126: I am sorry, but this part is not clear yet.  Conditional probability? Conditional on what? Is not that p is just an indicator function, that equals 1 when the i-th share and j-th share are in the l-th  and m-th slice respectively, otherwise equals zero? Please explain.

We edited this sentence as follows:  So the joint probability of both i and j to be placed in l and m divisions….

Reviewer 2 Report

The latest iteration of Afsharizand et al.'s paper represents a substantial improvement. The objects of study are all now well defined and the equations are all (almost) consistent. I have a few remaining comments that the authors should address before publication.

The authors never addressed one of my previous comments:
The first sentence is not a grammatically correct nor complete sentence. This should be remedied. It certainly is not a great first impression as it stands.

In Eq. (3), are the sine and cosine terms correct, or should they be swapped?

The caption for Fig. 3 seems misleading. It currently states that the figure shows "the return price and return volume of stocks".
However, my understanding is that the figure instead shows *the maximal alignment of pairs of stock vectors, when compared on the same day or with a one day lag*.
The caption should be modified to accurately describe what is being plotted.

The authors mention a "dendogram" several times. Do they mean "dendrogram" instead?
The former is not a known object.

Finally, a general comment to the authors:
Before submitting future papers, please do a much more thorough internal critical review of your manuscript. It is not fair to serious reviewers that invest significant time in evaluation---and not a good reputation to build for yourself---if the paper is of such poor quality upon its initial submission.

Author Response

The latest iteration of Afsharizand et al.'s paper represents a substantial improvement. The objects of study are all now well defined and the equations are all (almost) consistent. I have a few remaining comments that the authors should address before publication.

The authors never addressed one of my previous comments:

The first sentence is not a grammatically correct nor complete sentence. This should be remedied. It certainly is not a great first impression as it stands.

updated.

In Eq. (3), are the sine and cosine terms correct, or should they be swapped?

  done.

The caption for Fig. 3 seems misleading. It currently states that the figure shows "the return price and return volume of stocks".

However, my understanding is that the figure instead shows *the maximal alignment of pairs of stock vectors, when compared on the same day or with a one day lag*.

The caption should be modified to accurately describe what is being plotted.

  We have edited as follows: The maximal alignment of pairs of stock vectors, when compared on the same day or with a one day lag.

The authors mention a "dendogram" several times. Do they mean "dendrogram" instead? 

The former is not a known object.

 Thanks for the accurate point.  The “dendrogram” is correct.

Finally, a general comment to the authors:

Before submitting future papers, please do a much more thorough internal critical review of your manuscript. It is not fair to serious reviewers that invest significant time in evaluation---and not a good reputation to build for yourself---if the paper is of such poor quality upon its initial submission.

We greatly appreciate the reviewer with her/his constructive comments. Really, we can not compare the new version with the first version of the manuscript. We do not forget her/his important point.